# Rice Nudix Hydrolase OsNUDX2 Sanitizes Oxidized Nucleotides

**DOI:** 10.3390/antiox11091805

**Published:** 2022-09-13

**Authors:** Yuki Kondo, Kazuhide Rikiishi, Manabu Sugimoto

**Affiliations:** Institute of Plant Science and Resources, Okayama University, 2-20-1 Chuo, Kurashiki 710-0046, Okayama, Japan

**Keywords:** 8-oxo-dGTP, nudix hydrolase, *Oryza sativa*, transcriptional error, UV-C

## Abstract

Nudix hydrolase (NUDX) hydrolyzes 8-oxo-(d)GTP to reduce the levels of oxidized nucleotides in the cells. 8-oxo-(d)GTP produced by reactive oxygen species (ROS) is incorporated into DNA/RNA and mispaired with adenine, causing replicational and transcriptional errors. Here, we identified a rice *OsNUDX2* gene, whose expression level was increased 15-fold under UV-C irradiation. The open reading frame of the *OsNUDX2* gene, which encodes 776 amino acid residues, was cloned into *Escherichia coli* cells to produce the protein of 100 kDa. The recombinant protein hydrolyzed 8-oxo-dGTP, in addition to dimethylallyl diphosphate (DMAPP) and isopentenyl diphosphate (IPP), as did Arabidopsis AtNUDX1; whereas the amino acid sequence of OsNUDX2 had 18% identity with AtNUDX1. OsNUDX2 had 14% identity with barley HvNUDX12, which hydrolyzes 8-oxo-dGTP and diadenosine tetraphosphates. Suppression of the *lacZ* amber mutation caused by the incorporation of 8-oxo-GTP into mRNA was prevented to a significant degree when the *OsNUDX2* gene was expressed in *mutT*-deficient *E. coli* cells. These results suggest that the different substrate specificity and identity among plant 8-oxo-dGTP-hydrolyzing NUDXs and OsNUDX2 reduces UV stress by sanitizing the oxidized nucleotides.

## 1. Introduction

Ultraviolet (UV) radiation is a factor causing leaf chlorosis, necrosis, and plant growth retardation. Generation of reactive oxygen species (ROS) is one consequence of exposure to UV, leading to oxidative stress [1,2,3]. In fact, ROS attack DNA and produce oxidative nucleotides, such as 8-oxo-7,8-dihydro-2’-guanosine (8-oxo-G), known to be mutagenic [4]. 8-oxo-G-containing nucleotides, 8-oxo-dGTP and 8-oxo-GTP, which are incorporated into DNA as well as RNA, are mispaired with adenine to change genetic information [5,6,7]. To prevent the mutation, MutT in *Escherichia coli* and MTH1 in mammalian cells hydrolyze 8-oxo-(d)GTP to 8-oxo-(d)GMP [8,9,10,11,12,13,14,15].

Nudix hydrolases (NUDXs) hydrolyze various nucleotide diphosphate compounds linked to a moiety X to produce a nucleoside monophosphate and a phosphate linked to the moiety X [16]. NUDXs, which are widely distributed among microorganism, plants, and mammals, have a conserved nudix motif, GX_5_EX_7_REUXEEXGU, where U is usually Ile, Leu, or Val [16,17,18,19], and they are divided into subfamilies according to the preferred substrate. In plants, multiple genes of NUDXs have been identified from Arabidopsis and barley, in which there were 28 and 14 mRNAs, respectively [20,21]. Arabidopsis AtNUDX1 and barley HvNUDX12 hydrolyze 8-oxo-(d)GTP to 8-oxo-(d)GMP to prevent misincorporation into DNA/RNA [20,22,23]. The gene expression level of AtNUDX1 belonging to the 8-oxo-(d)GTP subfamily was not changed by paraquat, salinity, high light, or drought, which enhance the production of ROS; whereas that of HvNUDX12, belonging to the diadenosine tetraphosphates (Ap_4_A) subfamily, was upregulated by UV-C irradiation. These results suggest that the expression of *NUDX* genes encoding the 8-oxo-(d)GTP-hydrolyzing enzyme under UV irradiation differ among plant species.

UV irradiation of rice leaves significantly increased the genes of cyclobutane pyrimidine dimer photolyase, phenylalanine ammonialyase, lipoxygenase, chitinase, flavonoids, and total phenols, which are the enzymes and substances implicated in stress resistance [24,25,26]. However, there has been no report on the response of *NUDX* genes to UV stress and 8-oxo-dGTP-hydrolyzing NUDX in rice, whereas multiple rice *NUDX* (*OsNUDX*) genes encoding nudix motif-containing proteins have been predicted in databases. In order to find 8-oxo-dGTP-hydrolyzing OsNUDX, 17 putative OsNUDXs were assigned to subfamilies. The expression level of the rice *OsNUDX2* gene was increased 15-fold under UV-C irradiation in 17 rice *NUDX* genes. OsNUDX2 was classified in the dimethylallyl diphosphate (DMAPP)/isopentenyl diphosphate (IPP) subfamily. Moreover, it had low identities with AtNUDX1 and HvNUDX12. However, the purified OsNUDX2 hydrolyzed 8-oxo-dGTP in addition to DMAPP and IPP and, to a significant degree, prevented transcriptional errors in *E. coli* cells.

## 2. Materials and Methods

### 2.1. Plant Cultivation and UV Treatment

*Oryza sativa* L., Taichung 65 (T65), was used in this study. The seeds were germinated on filter paper that had been moistened with water at 20 °C in the dark. Three seedlings were transplanted in one Wagner pot (1/5000 a) filled with nutrient solution [27], and three replicate pots of each seedling were cultivated. After 14 days of cultivation under fluorescent light (150 μmol/m^2^/s) with a light/dark cycle of 20 h/4 h in a growth chamber to grow three leaves [25,26], the plants were irradiated with UV light (germicidal lamp GL-15, Panasonic, Tokyo, Japan) with an intensity of 200 μW/cm^2^, or they were irradiated with fluorescent light as a control. After exposure to UV light for 3 h, the plants were cultivated under fluorescent light for 2 weeks. The leaves for gene expression analysis were harvested after UV irradiation, frozen in liquid nitrogen, and stored at –80 °C.

### 2.2. qRT-PCR Analysis

Total RNA was isolated from leaves using the RNeasy Plant mini kit (Qiagen Inc., Tokyo, Japan) following the manufacturer’s instructions. Poly(A)^+^ RNA was purified from total RNA with the Poly (A) Purist MAG (Ambion Inc., Austin, TX, USA). Then, the purified poly(A)^+^ RNA was dissolved in the RNA storage solution. First-strand cDNA for quantitative RT-PCR was synthesized from poly(A) ^+^ RNA using a PrimeScript RT Master Mix (Takara Bio Inc., Shiga, Japan). Quantitative RT-PCR was performed in a mixture of 20 μL containing first-strand cDNA, SYBR Premix Ex Taq (Takara Bio Inc.), and 0.2 µmol of each primer combination (Appendix A), using LightCycler 2.0 (Roche Applied Science, Mannheim, Germany). The thermal cycle profile was 1 cycle of 95 °C for 10 s, followed by 40 cycles of 95 °C for 5 s and 60 °C for 20 s. The cDNA quantities of the respective genes were calculated using LightCycler 4.0 software (Roche Applied Science, Penzberg, Germany) and were normalized with that of the glyceraldehyde 3-phosphate dehydrogenase gene [28]. Expression analyses were conducted three times.

### 2.3. Expression and Purification of OsNUDX2

A predicted 2331 bp open reading frame of the *OsNUDX2* gene, which encodes 776 amino acid residues, was amplified with first-strand cDNA from the control leaves along with the primers 5′-GGGTACCGCGGAGCCGGAGGAGCGCCT-3′, which created a *Kpn* I site (denoted as underlined), and 5′-TTCTAGACTGATCGTTTGC-AAGCAGCT-3′, which created an *Xba* I site (denoted as underlined). The PCR product was cloned into the pGEM-T vector (Promega Corp., Madison, WI, USA). Then, the fragment of the plasmid digested by restriction enzymes was subcloned into a pCold I vector (Takara Bio Inc., Shiga, Japan), in which a polyhistidine tag gene was fused upstream from the start codon. The resulting plasmid, pOsNUDX2, was transformed into *E. coli* BL21 cells. Then, *E. coli* cells harboring pOsNUDX2 were grown at 37 °C in Luria–Bertani (LB) medium containing 50 μg/mL ampicillin. When the OD_600_ became 0.5, isopropyl-β-D-thiogalactopyranoside (IPTG) was added to the culture at a final concentration of 0.5 mM. After cultivation at 15 °C for 20 h for *E. coli* cells harboring pOsNUDX2, the cells were harvested by centrifugation and were frozen at −80 °C for at least 2 h. The frozen cell pellet was suspended in a protein extraction reagent (BugBuster™ HT; Merck and Co. Inc., Darmstadt, Germany), according to the manufacturer’s instructions. The resulting recombinant protein was purified using an Ni-NTA agarose column (Qiagen Inc., Venlo, Netherlands) initially equilibrated in 20 mM Tris-HCl buffer (pH 8.0) containing 0.5 M NaCl and 5 mM imidazole (Buffer A). The column was washed with Buffer A, followed by 60 mM imidazole in Buffer A. The absorbed protein was finally eluted with 1 M imidazole in Buffer A. The protein solution was dialyzed against 20 mM Tris-HCl buffer (pH 8.0) containing 1 mM DTT and 100 mM NaCl. The dialyzed solution was concentrated by an Ultracel YM-50 (Millipore Corp., Burlington, MA, USA).

### 2.4. Enzyme and Protein Assays

The hydrolytic activity of OsNUDX2 was assayed by quantitating the released free phosphate molecules using Malachite Green Phosphate Assay Kits (BioAssay Systems, Hayward, CA, USA). The reaction mixture (100 μL), which consisted of 50 mM Tris-HCl (pH 8.0), 5 mM MgCl_2_, 0.1 mM substrate, and the purified recombinant protein, was incubated at 37 °C for 30 min. The reaction mixture diluted with 50 mM Tris-HCl (pH 8.0) was mixed with the reagent of the kit, incubated for 30 min at 25 °C, and measured for absorbance at 620 nm. 8-oxo-dGTP, 8-oxo-dGDP, 8-oxo-dGMP (Jena Bioscience, Thuringia, Germany), and the product after reaction toward 8-oxo-dGTP were analyzed using HPLC with a Cosmosil C_18_ column (4.6 × 250 mm; Nacalai Tesque, Kyoto, Japan) equilibrated with 100 mM phosphate buffer (pH 6.0) and 5% (*v/v*) methanol, at a flow rate of 1.0 mL/min. The substrate and the reaction product were detected by absorption at 293 nm. The protein concentration was quantified according to Bradford [29] with bovine serum albumin Fraction V (Nacalai Tesque, Kyoto, Japan) as the standard. The assays were conducted three times.

### 2.5. Complementation Assay of Transcriptional Errors in mutT-Deficient E. coli

The open reading frame of the *OsNUDX2* gene was amplified with the forward primer, 5′-TGGAATTCGAGCCGGAGGAGCGCCTCG-3′, which created an *Eco*RI site (denoted as underlined), and the reverse primer, 5′-TTCTAGACTGATCGTTTGCAAGCAGCT-3′, which created an *Xba*I site (denoted as underlined). The PCR product was cloned into the pGEM-T vector, and the *Eco*RI- and *Xba*I-digested fragment of the plasmid was cut out and subcloned into an *Eco*RI- and *Xba*I-digested pTrc100 vector [23], obtaining a plasmid pTrc100/OsNUDX2. *E. coli* CC101 and *mutT*-deficient CC101T, which lacks the 8-oxo-dGTP hydrolase, harboring pTrc100 and pTrc100/OsNUDX2, were grown at 37 °C in an LB agar plate containing 50 µg/mL ampicillin, 1 mM IPTG, and 0.5 mg/mL 5-bromo-4-chloro-3-indolyl-β-D-galactoside (X-gal) overnight and kept at 25 °C for 2 days. β-Galactosidase activity was assayed using the colorimetric method as described by Miller [30]. *E. coli* cells were grown at 37 °C in an LB medium containing 50 µg/mL ampicillin and 1 mM IPTG. When the OD_600_ became 1.0, the culture was kept at 25 °C for 2 days. The cells were disrupted by sonication and were centrifuged to obtain the cell-free extract. After the cell-free extract was incubated with *o*-nitrophenyl-β-D-galactoside, the released *o*-nitrophenol was measured for absorbance at 420 nm. The assay was conducted three times.

## 3. Results and Discussion

### 3.1. Classification of Proteins Encoded by Putative Rice NUDX Genes

The genes, which are annotated as a nudix hydrolase, were sought using a rice annotation project database (RAP-DB https://rapdb.dna.affrc.go.jp, accessed on 8 October 2021) [31]; in total, 18 putative genes were identified. It was reported that there were 33 and 20 genes encoding NUDXs in rice [17,21].

RAP-DB offers highly reliable gene annotation based on the latest and accurate genome assembly including short-read sequences derived from next-generation sequencing. The differences in the number of *NUDX* genes might be caused by the accumulation of sequence data and the progress of the search engine. The gene fragment of each gene was amplified with primers designed from the sequence of the database, and 17 genes, from which the fragments were confirmed except for one gene (Os3g0810300), were assigned as *OsNUDX1-17* (Table 1) using the Clustal W program [32]. The deduced amino acid sequences of OsNUDX1-16 showed 47–89% identities with those of barley HvNUDXs and 42–68% identities with those of Arabidopsis AtNUDXs; whereas OsNUDX17, which is described as a NUDX domain-containing protein, showed no homologue of barley and Arabidopsis NUDXs. The alignment analysis of the amino acid sequences of OsNUDX1-17 indicated that the nudix motif was contained in the amino acid sequences of rice NUDXs (Figure 1A). The results showed that OsNUDX3, 6, 8, 12, 13, and 15 conserved the motif of GX_2_GX_6_G similar to Arabidopsis ApnA/ppGpp NUDX, OsNUDX5 and 14 conserved the motif of LLTXR[SA]X_3_RX_3_GX_3_FPGG similar to Arabidopsis CoA NUDX, and OsNUDX9 conserved the motif of SQX_2_WPXPXS similar to Arabidopsis NADPH NUDX. Based on these results, the OsNUDXs were classified to the following subfamilies: OsNUDX3, 6, 8, 12, 13, and 15 belong to the Ap_n_A/ppGpp NUDX; OsNUDX5 and 14 belong to the CoA NUDX; OsNUDX1 belongs to the ADP-ribose/NADH NUDX; OsNUDX2 belongs to the DMAPP/IPP NUDX; OsNUDX4 belongs to the GDP-mannose NUDX; OsNUDX7 belongs to the ADP-ribose/glucose NUDX; OsNUD10 belongs to the Thiamine NUDX; OsNUD11 belongs to the FAD NUDX; and OsNUDX16 belongs to the mRNA decapping enzyme. However, the rice NUDX genes encoding homologous to AtNUDX1 and HvNUDX12 that hydrolyze 8-oxo-dGTP were not identified. The MEME program [33], discovering ungapped motifs, identified three conserved motif compositions. All 17 OsNUDXs displayed a conserved nudix motif (Motif 1), and OsNUDX3, 6, 8, 12, 13, and 15 belonging to the Ap_n_A/ppGpp subfamily were characterized by Motif 2 and 3 (Figure 1B,C).

### 3.2. Expression of Rice NUDX Genes under UV-C Irradiation

The *HvNUDX12* gene was upregulated by UV-C irradiation [20]; then, the rice plants were exposed to UV-C irradiation, whose strength was not phytocidal, to identify the upregulated *OsNUDX* genes (Figure 2). The *OsNUDX* genes except *OsNUDX12* and *14* were detected under fluorescent light. There were 11 *OsNUDX* genes (*1–3*, *7–11*, *13–15*) upregulated significantly more than two-fold under UV-C irradiation, whereas the *OsNUDX4, 6, 12, 16, 17* genes showed no significant response to UV-C irradiation. In particular, the *OsNUDX2* and *11* genes showed expression levels of 195 and 161, representing increases, respectively, of 15-fold and 11-fold, which were classified to the DMAPP/IPP and FAD subfamilies; however, they had 18 and 13% identities to AtNUDX1 and 14 and 11% identities to HvNUDX12, respectively.

### 3.3. Purification and Enzymatic Characterization of Recombinant OsNUDX2

DMAPP and IPP are sources of plant terpenoid synthesis, which serves a broad range of physiological functions including defense and environmental sensing [34,35], and AtNUDX1 hydrolyzes not only 8-oxo-dGTP but also DMAPP and IPP [36], although OsNUDX2 had 18% identity with AtNUDX1. Therefore, OsNUDX2 was produced to evaluate its enzymatic property. The nucleotide sequence of the fragment amplified by PCR completely matched that of OsNUDX2 in the database. *E. coli* cells harboring pOsNUDX2 produced an extra protein, and the recombinant protein fused with a polyhistidine tag at its N-terminus was purified using Ni-NTA column chromatography. The molecular mass of OsNUDX2, estimated using SDS-PAGE, was ca. 100 kDa, which is similar to that calculated from the amino acid sequence of 90 kDa (Figure 3). The purified OsNUDX2 incubated with DMAPP and IPP showed production of phosphate, whose specific activities were, respectively, 183 ± 19 and 114 ± 3.1 nmol/min/mg. Furthermore, OsNUDX2 showed activity of 17.6 ± 2.7 nmol/min/mg for 8-oxo-dGTP, and the product after reaction was 8-oxo-dGDP and 8-oxo-dGMP (Figure 4), as did AtNUDX1. These results demonstrate that OsNUDX2 catalyzes the dephosphorylation of 8-oxo-dGTP.

OsNUDX2 and AtNUDX1 hydrolyzed DMAPP and IPP, and HvNUDX12 hydrolyzed Ap_4_A and guanoside-3’,5’-tetraphosphate in addition to the activity toward 8-oxo-dGTP. The identities of OsNUDX2 to AtNUDX1 and HvNUDX12 were 18 and 14%, respectively, and that between AtNUDX1 and HvNUDX12 was 13%. The *OsNUDX2* and *HvNUDX12* genes were induced by UV-C stress, whereas the expression of the *AtNUDX1* gene was not changed by treatment with paraquat, salinity, strong light, or drought, which cause ROS production. Therese results suggest the different substrate specificity and biological response among rice, Arabidopsis, and barley 8-oxo-dGTP-hydrolyzing NUDXs.

### 3.4. Prevention of Transcription Errors by OsNUDX2

To ascertain the ability of OsNUDX2 to eliminate the oxidized nucleotides in vivo, pTrc100/OsNUDX2 was introduced into *E. coli* CC101 and CC101T carrying an amber mutation in codon 461 of the *lacZ* gene, where the transversion mutation was reversed phenotypically to produce β-galactosidase [12,15]. *E. coli* CC101T cells harboring pTrc100, which lacked 8-oxo-dGTP hydrolase, produced blue colonies on the agar in the presence of X-gal. However, the formation of the blue colony was suppressed in *E. coli* CC101T cells harboring pTrc100/OsNUDX2 (Figure 5A). The β-galactosidase activities of the *E. coli* cells CC101T harboring pTrc100 and pTrc100/OsNUDX2 were, respectively, 2.78 ± 0.4 and 0.62 ± 0.3 nmol/min/mL-OD_600_, yielding the result that about 78% of the increase in activity was suppressed by the expression of OsNUDX2 (Figure 5B). These results demonstrated that OsNUDX2 sanitized 8-oxo-GTP to prevent the mutation caused by its incorporation into mRNA.

## 4. Conclusions

NUDXs hydrolyze various nucleotide diphosphate compounds and the response to abiotic and biotic stresses. This study demonstrated that rice OsNUDX2 hydrolyzed 8-oxo-dGTP, as did Arabidopsis AtNUDX1 and barley HvNUDX12, although OsNUDX2 showed 18% and 14% identities, respectively, and was classified into a different subfamily from AtNUDX2 and HvNUDX12. The expression of the *OsNUDX2* gene was induced by UV-C irradiation as *HvNUDX12*; whereas the expression of the *AtNUDX1* gene was not changed by oxidative stress. OsNUDX2 suppressed the *lacZ* amber mutation in *mutT*-deficient *E. coli* cells. Although the *OsNUDX2* gene was expressed under fluorescent light, and there was no evidence of the physiological function of OsNUDX2 in the rice seedling, the results suggest the different substrate specificity and biological response among rice, Arabidopsis, and barley 8-oxo-dGTP-hydrolyzing NUDXs and the OsNUDX2 responses to survive UV stress by sanitizing oxidized nucleotides.

## Figures and Tables

**Figure 1 antioxidants-11-01805-f001:**
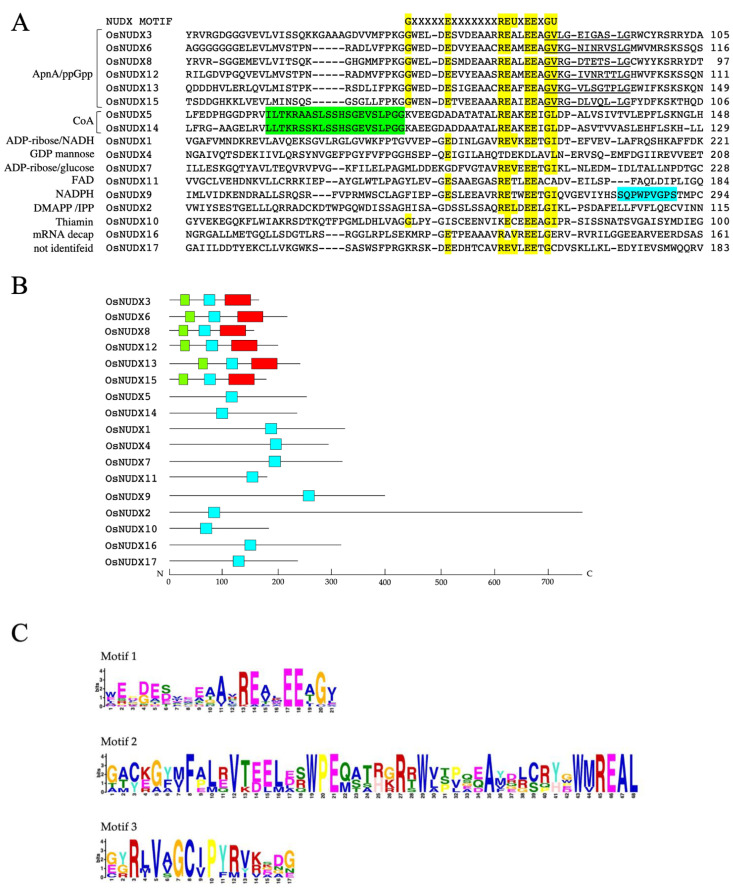
Alignment and conserved motifs of the amino acid sequences of the putative rice NUDXs. (**A**) Alignment was constructed by the CLUSTAL W program. Gaps, denoted by a dash, were introduced into the sequences to maximize the identity. The nudix motif is shown above the sequence. Identical amino acid residues to those of the nudix motif are shown as yellow boxes. The motifs of GX_2_GX_6_G in the ApnA/ppGpp NUDX, LLTXR[SA]X_3_RX_3_GX_3_FPGG in the CoA NUDX, and SQX_2_WPXPXS in the NADPH NUDX are shown, respectively, as underlined, green boxes, and blue boxes. The deduced subfamily is shown at the left side; (**B**) distribution of conserved motif 1 (blue boxes), motif 2 (red boxes), and motif 3 (green boxes) in the OsNUDXs predicted by the MEME program. The x-axis indicates the length of the amino acid residues from the N-terminal (N) to the C-terminal (C); (**C**) the conserved protein motifs in OsNUDXs. The y-axis indicates the relative frequency of each amino acid residue among the OsNUDXs.

**Figure 2 antioxidants-11-01805-f002:**
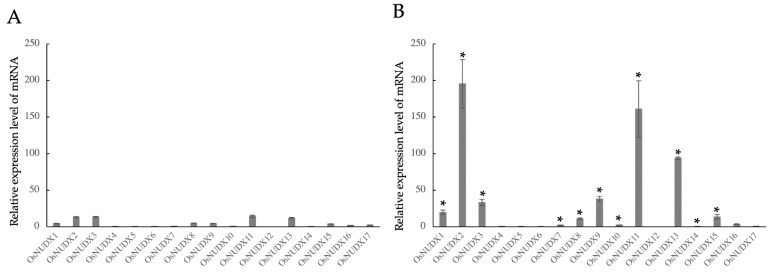
Expression profiles of the *OsNUDX* genes under UV-C irradiation. Total RNAs isolated from rice leaves (**A**) before and (**B**) after UV-C irradiation were subjected to quantitative RT-PCR. The expression levels were normalized with that of the glyceraldehyde 3-phosphate dehydrogenase gene as an internal control. The values denote the means ± SD (*n* = 3). * *p* < 0.01.

**Figure 3 antioxidants-11-01805-f003:**
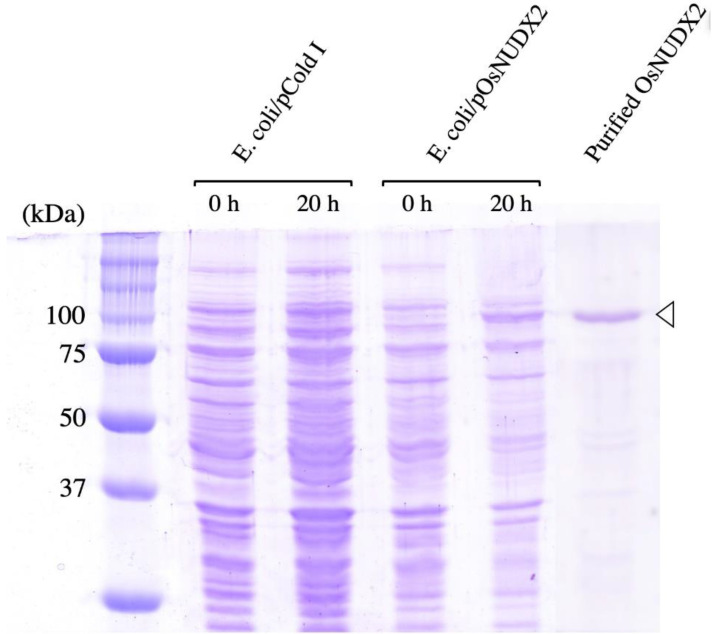
Production and purification of OsNUDX2 in *E. coli* cells using SDS-polyacrylamide gel. Total cell lysates of *E. coli* cells harboring pCold I and pOsNUDX2, before and after IPTG induction for 20 h, and purified OsNUDX2 were electrophoresed using 10% SDS-PAGE with the molecular mass marker series for calibration, with detection by Coomassie Brilliant Blue R-250 staining. An arrowhead indicates the OsNUDX2 protein.

**Figure 4 antioxidants-11-01805-f004:**
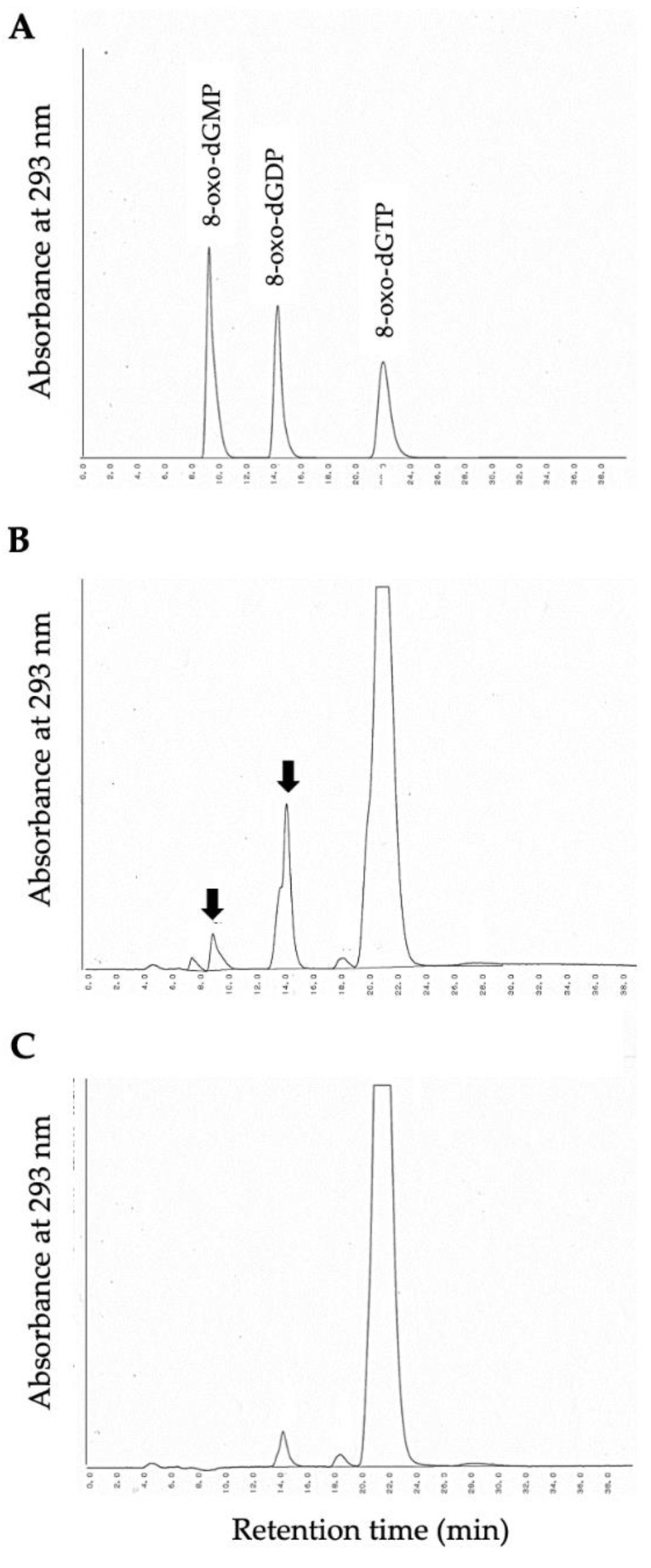
Identification of 8-oxo-dGTP degradation products by OsNUDX2. The reaction mixture containing 8-oxo-dGTP in the absence or presence of purified OsNUDX2 was incubated at 37 °C for 30 min and was subjected to HPLC. (**A**) Elution profiles of standard 8-oxo-dGTP, 8-oxo-dGDP, and 8-oxo-dGMP; (**B**) the elution profile of the reaction mixture with OsNUDX2; and (**C**) the elution profile of the reaction mixture without OsNUDX2.

**Figure 5 antioxidants-11-01805-f005:**
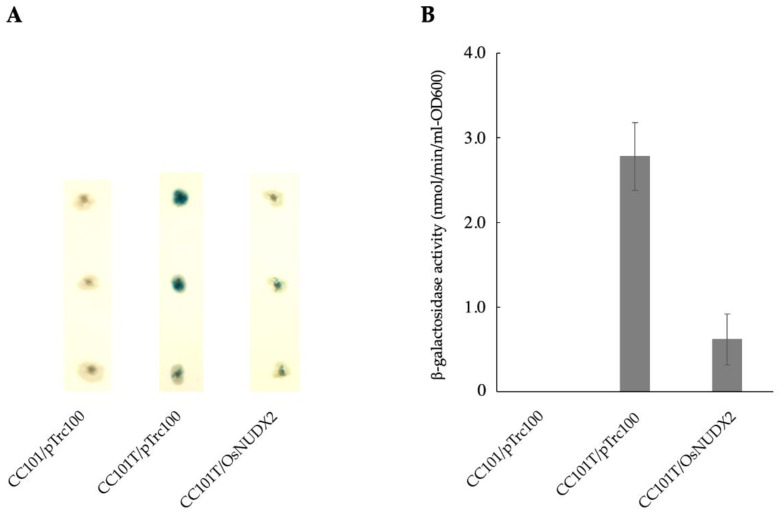
Suppression of transcription errors by the expression of OsNUDX2. (**A**) *E. coli* cells CC101 harboring pTrc100, CC101T harboring pTrc100, and pTrc100/OsNUDX2 were grown on an LB medium plate containing X-gal and (**B**) the β-galactosidase activity in the cell-free extracts of *E. coli* cells CC101 harboring pTrc100, CC101T harboring pTrc100, and pTrc100/OsNUDX2. The values denote the means ± SD (*n* = 3).

**Table 1 antioxidants-11-01805-t001:** Putative *NUDX* genes in the rice genome and identification of *OsNUDX* genes with those of *HvNUDX* and *AtNUDX* genes.

Gene (Deduced Subfamily)	Locus ID	Position	Orthologue	(Subfamily)	Identity (%)
*OsNUDX1*	Os06g0634300	chr06:25718152..25722139 (−strand)	HvNUDX1	(ADP ribose/NADH)	82
(ADP-ribose/NADH)			AtNUDX2	(ADP ribose/NADH)	57
			AtNUDX7	(ADP ribose/NADH)	47
*OsNUDX2*	Os02g0793300	chr02:33710064..33719486 (+strand)	HvNUDX2		89
(DMAPP/IPP)			AtNUDX3	(DMAPP/IPP)	68
*OsNUDX3*	Os06g0255400	chr06:8069204..8069983 (−strand)	HvNUDX3	(ApnA/ppGpp)	75
(ApnA/ppGpp)			HvNUDX8	(ApnA/ppGpp)	61
			AtNUDX18		44
			AtNUDX17		43
*OsNUDX4*	Os05g0117500	chr05:922037..926090 (+strand)	HvNUDX4	(GDP mannose)	74
(GDP-mannose)			AtNUDX9	(GDP mannose)	59
*OsNUDX5*	Os05g0209400	chr05:6766494..6768972 (+strand)	HvNUDX5	(CoA)	68
(CoA)			HvNUDX14	(CoA)	49
			AtNUDX11	(CoA)	46
			AtNUDX15	(CoA)	46
*OsNUDX6*	Os04g0399300	chr04:19735871..19739665 (+strand)	HvNUDX6	(ApnA/ppGpp)	58
(ApnA/ppGpp)			AtNUDX12		43
			AtNUDX13	(APnA/ppGpp)	42
*OsNUDX7*	Os06g0129700	chr06:1574614..1577176 (−strand)	HvNUDX7	(ADP ribose/glucose)	80
(ADP-ribose/glucose)			AtNUDX14	(ADP ribose/glucose)	59
*OsNUDX8*	Os02g0734300	chr02:30625668..30626865 (+strand)	HvNUDX8	(ApnA)	78
(ApnA)			HvNUDX3	(ApnA)	66
			AtNUDX18		52
			AtNUDX17		49
			AtNUDX21		48
*OsNUDX9*	Os06g0141166	chr06:2157903..2160576 (−strand)	HvNUDX9	(NADPH)	79
(NADPH)			AtNUDX19	(NADPH)	55
*OsNUDX10*	Os09g0322200	chr09:9379558..9383831 (−strand)	HvNUDX10	(Thiamine)	79
(Thiamine)			AtNUDX20	(Thiamine)	63
			AtNUDX24		60
*OsNUDX11*	Os09g0553300	chr09:21922941..21926057 (−strand)	HvNUDX11	(FAD)	72
(FAD)			AtNUDX23	(FAD)	49
*OsNUDX12*	Os02g0520100	chr02:18936409..18939517 (−strand)	HvNUDX6	(ApnA/ppGpp)	59
(ApnA/ppGpp)			AtNUDX12		48
			AtNUDX13	(ApnA/ppGpp)	48
*OsNUDX13*	Os11g0531700	chr11:19353108..19355768 (+strand)	HvNUDX6	(ApnA/ppGpp)	47
(ApnA/ppGpp)			AtNUDX12		46
			AtNUDX13	(ApnA/ppGpp)	46
*OsNUDX14*	Os08g0375900	chr08:17634789..17639334 (+strand)	HvNUDX14	(CoA)	70
(CoA)			AtNUDX15	(CoA)	56
			AtNUDX11	(CoA)	53
*OsNUDX15*	Os07g0212300	chr07:6137057..6141022 (−strand)	AtNUDX16	(ApnA/ppGpp)	62
(ApnA/ppGpp)			AtNUDX13	(ApnA/ppGpp)	44
			AtNUDX12		42
*OsNUDX16*	Os02g0805900	chr02:34378712..34382107 (−strand)	AtDCP2	(mRNA decap)	58
(mRNA decap)					
*OsNUDX17*	Os06g0712200	chr06:30117755..30118750 (+strand)	not identified		

## Data Availability

The data are contained within the article and supplementary materials.

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
