# Peer review of "Rice Nudix Hydrolase OsNUDX2 Sanitizes Oxidized Nucleotides"

_antioxidants, 2022, doi:10.3390/antiox11091805_

Round 1

Reviewer 1 Report

In this MS, the authors described identified a rice OsNUDX2 gene, of which the expression level was increased 15-fold under UV-C irradiation. The recombinant protein hydrolyzed 8-oxo-dGTP in addition to dimethylallyl diphosphate (DMAPP) and isopentenyl diphosphate (IPP). Suppression of the lacZ amber mutation caused by the incorporation of 8-oxo-GTP into mRNA was prevented to a significant degree when OsNUDX2 gene was expressed in a mutT deficient Escherichia coli cells. These results suggest that the different substrate specificity and identity among plant 8-oxo-dGTP-hydrolyzing NUDXs and OsNUDX2 reduces UV stress by sanitizing the oxidized nucleotides. There are some comments should be addressed as follows:

(1)   Table 1 should be moved to the supplemental data part;

(2)   Add a figure containing the domains information of this big family;

(3)   Change Table 3 into a bar chart to show the expression level.

Author Response

Comment 1.  Table 1 should be moved to the supplemental data part.

>Answer.  According to your comment, Table 1 was removed to supplemental materials as Table S1.

Commnets 2. Add a figure containing the domains information of this big family.

>Answer.  According to your comment, we added the figures showing domains in 17 OsNUDXs as Figure 1B and C and described in the text. (Page 5 Line 173-176)

Comment 3. Change Table 3 into a bar chart to show the expression level.

>Answer.  According to your comment, Table 3 changed into a bar chart as Figure 2.

Reviewer 2 Report

Review of the manuscript entitled “Rice Nudix Hydrolase OsNUDX2 Induced under UV-C Irradiation Sanitizes Oxidized Nucleotides” by Yuki Kondo, Kazuhide Rikiishi and Manabu Sugimoto.

General comments

The manuscript presented research work focusing on a predicted gene encoding a nudix hydrolase, designated OsNUDX2, in rice. A total of 18 predicted nudix genes were selected from the rice genome and quantified for transcript levels in samples of early seedlings from the T65 grown in a UV treatment and a control environment. OsNUDX2 was one of the genes different in the transcription level between the UV stress and control conditions. Thus, OsNUDX2 was expressed in E. coli and the recombinant protein assayed for hydrolytic activities of the oxidized nucleotide 8-oxo-dGTP. OsNUDX2 was also cloned into an E. coli mutant for the 8-oxo-dGTP hydrolase gene and evaluated for its complementation effect in the bacteria. The transcription data from the rice seedlings and biochemical and genetic (complementation) data from the bacteria support the implication that OsNUDX2 is likely involved in the regulation of resistance to the abiotic stress by UV irradiation in the rice plant. These data are preliminary toward understanding the nudix gene family in plants and would be interested by researchers working on areas of antioxidants and functional genomics. This manuscript needs significant work to improve writing in English and scientific/logical expressions, including the title, abstract and tables.

Specific comments

1. About the title: It is difficult to understand authors’ point. Correct grammatic errors and outline the discovery straightforward.

Introduction:

2. The first paragraph and abstract: Rephrase this paragraph and statements in the abstract to state clearly the cause-effect relation from a UV stress to ROS, nucleotide oxidation and potential mutation. Not clear if MutT and MTH1 are proteins or cell lines.  

3. The second paragraph: Start with the definition of NUDX before a review of the information from Arabidopsis and barley. The last sentence is not clear. 

4. The third (last) paragraph: Add 1) statements to define the objective(s) of this research; and 2) a brief review on the background information about nudix genes and research on the UV stress in rice. These lines of information are important to understand the rationale and significance of the research and help organize the following sections as well.    

Materials and methods

5. 2.1: start with the genotype/variety of rice; then move to the source of seeds and plant cultivation. Seedling of 14 d old are sensitive to the environmental stress. Refer to a reference of rice physiology to help justify the experimental design. It is suggested to repeat the treatment with older seedlings.  

6. 2.2: Table 1. 1) change the title of the first column from gene to locus ID or add a column to show the loci; and 2) add physical positions of the sequences on the reference genome of rice. 7.

7. 2.3: Add information on the size and structure of the hydrolase gene at locus Os02g0793300 (>20 exons) to know better of the protein.  

Results

8. 3.1: Move some of the background information to the introduction; follow conventions to present names of genes and proteins; rephrase the title of Table 2 to indicate that this is a summary on predicted genes for NUDX in the rice genome and also indicate if the prediction was based on their similarities with those from Arabidopsis or barley; change the title of the column “Gene ID” to Locus ID; and replace the title “homologue” with “orthologue”.

9. 3.2: Add a column to Table 3 to indicate the statistic for the comparisons. Indicate the internal control to make sense of the fold change.

10. Figure 2. The molecular weight of OsNUDX2 could be better estimated using 6 or 8 % SDS-PAGE. 

Conclusions

11. It is suggested to replace “conclusions” with discussion and limit the discussion in the bacterial system.

12. Correct grammatical, logical, or biological errors throughout the manuscript. For example, in the leading statement in the section “NUDXs have diverse substrate specificity and response to abiotic and biotic stress”, it is supposed that the expressions “response” and “stress” are plural, and it is unclear about “diverse substrate specificity”. The data shown were for complementation and it is not appropriate to annotate as “suppression”.     

Author Response

Comment 1. About the title: It is difficult to understand authors’ point. Correct grammatic errors and outline the discovery straightforward.

>Answer.  According to your comment, title was modified. (Page 1, Line 2-3)

Comment 2. The first paragraph and abstract: Rephrase this paragraph and statements in the abstract to state clearly the cause-effect relation from a UV stress to ROS, nucleotide oxidation and potential mutation. Not clear if MutT and MTH1 are proteins or cell lines. 

>Answer. According to your comment, we modified the sentences in abstract and the first paragraph in Introduction. (Page 1 Line 9-10, Line 29-31)

Comment 3. The second paragraph: Start with the definition of NUDX before a review of the information from Arabidopsis and barley. The last sentence is not clear. 

>Answer. According to your comment, we showed the definition of NUDX first and modified the last sentence to clear the point that the expression of NUDX genes encoding 8-oxo-(d)GTP-hydrolyzing enzyme under UV irradiation differ among plant species. (Page 1 Line 32-45)

Comment 4. The third (last) paragraph: Add 1) statements to define the objective(s) of this research; and 2) a brief review on the background information about nudix genes and research on the UV stress in rice. These lines of information are important to understand the rationale and significance of the research and help organize the following sections as well.   

>Answer. According to your comment, we added the statements to the objective of this research, a background information after 2nd paragraph. (Page 2 Line 46-50).

Comment 5. 2.1: start with the genotype/variety of rice; then move to the source of seeds and plant cultivation. Seedling of 14 d old are sensitive to the environmental stress. Refer to a reference of rice physiology to help justify the experimental design. It is suggested to repeat the treatment with older seedlings.

>Answer. According to your comment, the genotype and variety of rice we used was described in the first sentence. (Page 2 Line 60)

The leaves of seedings of 14 d old are large enough to receive the light evenly, and there are many reports of experiments that leaves of young plants are used to analyze the UV-responsive genes and proteins in rice.

Comment 6. 2.2: Table 1. 1) change the title of the first column from gene to locus ID or add a column to show the loci; and 2) add physical positions of the sequences on the reference genome of rice.

>Answer. According to your comment, gene was changed to Locus ID and physical positions was added in the title of the first column. (Table 1)

Comment 7. Add information on the size and structure of the hydrolase gene at locus Os02g0793300 (>20 exons) to know better of the protein.

>Answer. According to your comment, the gene size of open reading frame and the length of amino acid residues of OsNUDX2 at locus Os02g0793300 was shown in the text. (Page 2 Line 88-89)

Comment 8. 3.1: Move some of the background information to the introduction; follow conventions to present names of genes and proteins; rephrase the title of Table 2 to indicate that this is a summary on predicted genes for NUDX in the rice genome and also indicate if the prediction was based on their similarities with those from Arabidopsis or barley; change the title of the column “Gene ID” to Locus ID; and replace the title “homologue” with “orthologue”.

>Answer. The background information described in 3.1 was moved to the Introduction. (Page 1 Line 34-37)

The title of Table 1 was edited according to your comment. Gene ID and Homologue were changed to Locus ID and Orthologue, respectively. (Table 1)

Comment 9. 3.2: Add a column to Table 3 to indicate the statistic for the comparisons. Indicate the internal control to make sense of the fold change.

>Answer. Table 3 changed into a bar chart as Figure 2 according to Reviewer #1 and the statistic was indicated in Figure 2 according to your comment. (Figure 2)

Comment 10. The molecular weight of OsNUDX2 could be better estimated using 6 or 8 % SDS-PAGE.

>Answer. Thank you for your comment. We mistyped the gel concentration. The gel concentration we used is 10% (Figure 3 legend line 3). As shown in Figure 3, an extra protein can be recognized in the lane of sample E. coli/pOsNUDX2 for 20 h. A standard curve can be created with maker proteins separated in this gel.

 Comment 11. It is suggested to replace “conclusions” with discussion and limit the discussion in the bacterial system.

>Answer. The paragraph described about the identity and expression of OsNUDX2 in 3.4 was moved to 3.3.  3.4 limited the results and discussion about sanitizing oxidized nucleotide by OsNUDX2 in the bacterial system. (Page 8 Line 208- Page 9 Line 227)

Comment 12. Correct grammatical, logical, or biological errors throughout the manuscript. For example, in the leading statement in the section “NUDXs have diverse substrate specificity and response to abiotic and biotic stress”, it is supposed that the expressions “response” and “stress” are plural, and it is unclear about “diverse substrate specificity”. The data shown were for complementation and it is not appropriate to annotate as “suppression”. 

>Answer. We corrected errors and replaced “diverse substrate specificity” with “various nucleotide diphosphate compounds” as you pointed out (Page 9 Line 230-231). 

As shown in the subtitle 2.4, this is a “complementation” assay in E. coli cells, but the results indicate that OsNUDX2 suppressed the transcriptional errors.

Round 2

Reviewer 1 Report

All of the comments have been replied. 

Author Response

Authors thank your comment.

We checked and corrected scientific or grammatical errors in the manuscript. 

Reviewer 2 Report

Specific comments

1. Title: Improve the title: 1) to reflect the hydrolase is only part of a putative sanitizing pathway, and 2) add void misleading by addition of a phrase, like “in E. coli”.

2. Abstract: Improve the abstract for clarity and succinctness. Authors may try: 1) use of simple sentences to express key information/results correctly in English; 2) Start with the function of the hydrolase, instead of ROS; and 3) add necessary information to define the recombinant protein”.

3. Introduction: 1) Correct scientific or grammatical errors in expressions. For example, it is “stress by UV radiation”, not any “UV radiation” that causes the problem’; and it is the proteins, not the protein-coding genes that belong a subfamily; and “there is no report”. 2) add a short summary on predicted NUDX genes in the annotated reference genome of rice and indicate if there is information the response of the gene family to the other stress factors. And 3) Clearly define the objectives of this research to help evaluate the importance or implications of the listed results and present and organize the following sections.   

Materials and methods

4. 1) use a citation to replace the information on the culture solution; 2) add information on the sample size and number of biological replicates (figure 2) used for the experiment; 3) methods and parameters used to quantify responses of the seedlings to the UV stress; and 4) cite papers to support that it is reasonable to use 14-d seedlings for stress analysis.

5. It is not clear that the coding sequence of OsNUDX2 was predicted based on cDNAs from Nipponbare or T65. Add information on the similarity or difference in the selected gene between the cultivars.

Results

6. Change the subtitle 3.1 and the following annotations to reflect the analysis of predicted genes in the rice genome. Data from this research may be used to correct or improve annotations to the reference genome sequence in the database.

Conclusions

7. Correct some statements as: 1) the transcript of gene was also detected in the control samples (Figure 2) and 2) this research did not provide direct evidence on the gene’s function in the rice seedling.

Author Response

The authors thank the comments from reviewer.  According to the comments, the authors revised the manuscript as follows:

Comment 1. Title: Improve the title: 1) to reflect the hydrolase is only part of a putative sanitizing pathway, and 2) add void misleading by addition of a phrase, like “in E. coli

>Answer.   This research is to find a NUDX from rice that sanitizes 8-oxo-dGTP.  Enzyme assay (in vitro experiment) and complementation assay using E. coli cells (in vivo experiment) are the way to show that OsNUDX2 sanitizes 8-oxo-dGTP. From these results and according to your previous comment, we modified the title to show the discovery straightforward.

Comment 2. Abstract: Improve the abstract for clarity and succinctness. Authors may try: 1) use of simple sentences to express key information/results correctly in English; 2) Start with the function of the hydrolase, instead of ROS; and 3) add necessary information to define the recombinant protein”.

>Answer. According to your comment, we modified the sentences and add the information of the recombinant protein in abstract. (Page 1 Line 8-13)

Comment 3. Introduction: 1) Correct scientific or grammatical errors in expressions. For example, it is “stress by UV radiation”, not any “UV radiation” that causes the problem’; and it is the proteins, not the protein-coding genes that belong a subfamily; and “there is no report”. 2) add a short summary on predicted NUDX genes in the annotated reference genome of rice and indicate if there is information the response of the gene family to the other stress factors. And 3) Clearly define the objectives of this research to help evaluate the importance or implications of the listed results and present and organize the following sections.   

>Answer. According to your comment, we corrected the error parts (Page 1 Line 25, Line 41, Page 2 Line 50), added the information of predicted NUDX gene (Page 2 Line 51-52), and modified the sentences to clear the objective (Page 2 Line 52-59).

Comment 4. 1) use a citation to replace the information on the culture solution; 2) add information on the sample size and number of biological replicates (figure 2) used for the experiment; 3) methods and parameters used to quantify responses of the seedlings to the UV stress; and 4) cite papers to support that it is reasonable to use 14-d seedlings for stress analysis.

>Answer. According to your comment, we showed 1) a citation and eliminated the component of the culture solution, 2) the information about sample size and number of biological replicates, 3) parameters of UV stress, and 4) cited papers. (Page 2 Line 64-68)

Comment 5. It is not clear that the coding sequence of OsNUDX2 was predicted based on cDNAs from Nipponbare or T65. Add information on the similarity or difference in the selected gene between the cultivars.

>Answer. The nucleotide sequence of the fragment amplified by PCR matched that of OsNUDX2 in database completely. According to your comment, the sentence was added (Page 5 Line 195-196).

Comment 6. Change the subtitle 3.1 and the following annotations to reflect the analysis of predicted genes in the rice genome. Data from this research may be used to correct or improve annotations to the reference genome sequence in the database.

>Answer. Authors thank your comment. The subtitle 3.1 was changed to indicate the classification of rice NUDX proteins. (Page 3 Line 146)

Comment 7. Correct some statements as: 1) the transcript of gene was also detected in the control samples (Figure 2) and 2) this research did not provide direct evidence on the gene’s function in the rice seedling.

>Answer. According to your comment, the sentence “OsNUDX2 gene was expressed under the fluorescent light and there is no evidence of physiological function of OsNUDX2 in the rice seedling” was added in the Conclusions. (Page 9 Line 240-241)